# Influenza D Virus: A Review and Update of Its Role in Bovine Respiratory Syndrome

**DOI:** 10.3390/v14122717

**Published:** 2022-12-05

**Authors:** Miguel Ruiz, Andrea Puig, Marta Bassols, Lorenzo Fraile, Ramon Armengol

**Affiliations:** 1Department of Animal Science, ETSEA, University of Lleida, 25198 Lleida, Spain; 2Agrotecnio Research Center, ETSEA, University of Lleida, 25198 Lleida, Spain

**Keywords:** bovine respiratory disease, cattle, etiopathogeny, influenza D virus, zoonosis

## Abstract

Bovine respiratory disease (BRD) is one of the most prevalent, deadly, and costly diseases in young cattle. BRD has been recognized as a multifactorial disease caused mainly by viruses (bovine herpesvirus, BVDV, parainfluenza-3 virus, respiratory syncytial virus, and bovine coronavirus) and bacteria (*Mycoplasma bovis*, *Pasteurella multocida*, *Mannheimia haemolytica* and *Histophilus somni*). However, other microorganisms have been recognized to cause BRD. Influenza D virus (IDV) is a novel RNA pathogen belonging to the family *Orthomyxoviridae*, first discovered in 2011. It is distributed worldwide in cattle, the main reservoir. IDV has been demonstrated to play a role in BRD, with proven ability to cause respiratory disease, a high transmission rate, and potentiate the effects of other pathogens. The transmission mechanisms of this virus are by direct contact and by aerosol route over short distances. IDV causes lesions in the upper respiratory tract of calves and can also replicate in the lower respiratory tract and cause pneumonia. There is currently no commercial vaccine or specific treatment for IDV. It should be noted that IDV has zoonotic potential and could be a major public health concern if there is a drastic change in its pathogenicity to humans. This review summarizes current knowledge regarding IDV structure, pathogenesis, clinical significance, and epidemiology.

## 1. Introduction

Bovine respiratory disease (BRD) is the most important multifactorial disease affecting young cattle. It is the leading cause of mortality and post-weaning morbidity, with a significant impact on the reduction of production performance and increase of veterinary cost, affecting the economic balance of cattle farms worldwide. In addition, it decreases animal welfare and increases the use of antibiotics [1,2,3,4]. The occurrence of BRD is usually triggered by the presence of one or more viruses and/or bacteria that are favored by a set of individual and environmental factors [3]. Additionally, the lung anatomy of cattle has the singularity of being small in proportion to the body size of the animal, which makes it more susceptible to respiratory diseases.

The environmental predisposing factors that most frequently contribute to the onset of BRD are weaning stress, sudden change of feeding, transportation, overcrowding, grouping animals from different origins and ages, poor ventilation, wind drafts, and high relative humidity [2,5]. The pathogens most frequently involved in BRD are bovine syncytial respiratory virus (BRSV), bovine parainfluenza type 3 virus (PI3V), bovine herpesvirus type 1 (BoHV-1), bovine viral diarrhea virus (BVDV), bovine coronavirus (BCoV), *Mycoplasma bovis* (*M. bovis*), *Pasteurella multocida* (*P. multocida*), *Mannheimia haemolytica* (*M. haemolytica*), and *Histophilus somni* (*H. somni*) [6]. Respiratory viruses can reach the lower respiratory tract and induce disease on their own, or they may cause immunosuppression, damaging the respiratory epithelium and the defense mechanisms of the respiratory tract. Both pathogenesis mechanisms may favor the multiplication and subsequent colonization of the lower respiratory tract by secondary bacterial species, some of which may be present in the nasopharynx [3,4,6] (Figure 1). The complexity of this pathology is often reinforced by the presence of mixed infections (coinfections) involving viruses and bacteria [6]. Together with the control of risk factors, numerous commercial vaccines consisting of combinations of bacteria and live attenuated or inactivated viruses are widely used to prevent BRD. Nevertheless, the incidence of BRD has increased in recent decades [7].

Bovines were not considered susceptible to influenza viruses until the discovery of influenza D virus (IDV). This new species of RNA virus of the *Orthomyxoviridae* family has been identified as a new etiological agent involved in BRD since 2011, and studies to date show that this virus may play a role in the appearance of BRD [3]. Since 2011, the interest in research on IDV has not stopped increasing. This highlights the importance and global impact that this new virus may have, which mainly affects cattle, although there is a wide range of other species that can act as hosts.

The main aim of this work is to review and update all the information regarding IDV that affects cattle, detailing what is known and what remains to be clarified about this virus and its role in BRD.

## 2. Materials and Methods

To prepare the complete review of the subject matter, the following databases of scientific articles and specialized books were used: Google Scholar, PubMed, Scopus, and Web of Science (accessed from November 2021 to October 2022). In the search tool of each database, a general review of the IDV in cattle was first carried out using combinations of keywords with the Boolean operators AND, OR, and NOT. These searches were then refined with more specific keywords linked to the topic in each of the 4 scientific databases used (Table 1).

## 3. Results and Discussion

### 3.1. Origin of IDV and Its Role in the BRD

The first time that IDV was isolated was in April 2011 in the state of Oklahoma (U.S.A.) in a pig with clinical signs similar to those of swine flu [8], and subsequently isolated several times in respiratory samples of bovines from different parts of the world, being a virus with potential for zoonotic and interspecific transmission [9]. Some studies of stored sera reported that IDV had been circulating in the cattle population of the United States (U.S.A.) long before it was identified as a new virus. Thus, available information suggests that the circulation could go back to 2003 [10]. Note that all articles related to IDV in cattle are relatively recent, since the oldest article was published in 2014.

Influenza viruses, commonly known as “flu viruses”, are contagious pathogenic microorganisms that belong to the *Orthomyxoviridae* family, which consists of four species: influenza A virus (IAV), influenza B virus (IBV), influenza C virus (ICV), and influenza D virus (IDV). In August 2016, the International Committee on Taxonomy of Viruses (ICTV) officially classified IDV as a new species that belongs to the genus *Deltainfluenzavirus*, in spite of some genomic similarity to ICV. Regarding their morphology, IDVs present an envelope and a segmented genome, composed of seven negative-sense, single-stranded RNA segments, and lacking neuraminidases, as does ICV. Virions are 80–120 nanometers in diameter [11] (Figure 2).

The IDV that was first identified shared approximately 50% homology with the human influenza C virus (ICV) and was initially thought to be a new subtype of ICV, suggesting a common ancestor for both viruses. However, IDV lacked cross-reactivity against human ICV antibodies, so it was classified into a new genus in the *Orthomyxoviridae* family [11]. It was also pointed out that IDV could be derived from human ICV, based on their similarities in terms of genetic sequences [12]. IDV has a broader cellular and host tropism than ICV, which could be attributed to the hemagglutinin-esterase fusion (HEF) glycoproteins of IDV having an open receptor-binding cavity to house various cellular receptor molecules [12,13,14].

The HEF glycoproteins aid virus entry into host cells, as well as being the main target of neutralizing antibodies generated during IDV infection. In addition, some studies highlight that IDV has exceptional stability at high temperatures and acidity due to the role of HEF glycoproteins, and it is considered the most stable of the four influenza viruses [11]. Thus, IDV could resist and retain its infectivity even after exposure to a temperature of 53 °C for 120 min. Furthermore, IDV only lost 20% of the original infectivity when subjected to a low pH of 3.0 for 30 min, compared with the rest of influenza viruses that were completely inactive at low pH [12].

Since 2011, the complete genomes of more than fifty cattle and five pig IDV strains have been sequenced in six countries: U.S.A., France, Italy, Ireland, Japan, and China. Phylogenetic analyses revealed that IDV has been circulating globally, with at least four distinct genetic and antigenic lineages [11,15]:D/swine/Oklahoma/1334/2011 (D/OK). Detected in Europe (France, Italy, and Ireland), America (U.S.A. and Mexico), and Asia (China);D/bovine/Oklahoma/660/2013 (D/660). Detected in Europe (Italy) and America (U.S.A. and Mexico);D/bovine/Yamagata/10710/2016 (D/Yama2016). Detected in Asia (Japan);D/bovine/Yamagata/1/2019 (D/Yama2019). Detected in Asia (Japan and China).

It is important to highlight that D/OK and D/660 frequently showed rearrangement events with each other and antigen–antibody cross-reactivity between them. Some studies reveal that the strains reported in the bovine population of Japan were part of a single group that was distinct from the strains reported in other countries. These Japanese IDV strains diverged substantially from the D/OK and D/660 lineages, raising the possibility of an evolution and unique pathology of IDV as two lineages [11,12]. Research continues to report new possible lineages worldwide. Recently, in the first molecular detection of IDV in South America from a case of BRD, it was found that Brazilian IDV circulating in the cattle herds was phylogenetically divergent from known IDVs described in North America, Europe, and Asia [16]. Additionally, new lineages have been reported in Turkey [17] and Namibia [18].

The bovine species was not considered susceptible to influenza viruses until the discovery of IDV; this virus has been designated as a new etiological agent of BRD, and studies to date show it may play a role in respiratory disease [3,16]. Experimentally, in a controlled environment and in the absence of coinfecting pathogens, BRD alone caused mild to moderate respiratory disease [4]. In addition, IDV has been shown to be highly transmissible and to predispose transmission or potentiate the effects of other respiratory pathogens involved in BRD [4,16,19].

To identify the viruses associated with BRD, a metagenomic analysis was performed in California (U.S.A.) in suckling calves 27–60 days of age with clinical signs of acute BRD (n = 50) and in healthy/asymptomatic calves (n = 50). Samples were collected using deep nasopharyngeal swabs (DNS). The viral metagenomic sequencing showed that IDV was a commonly identified microorganism and significantly associated with calves diagnosed with BRD [20]. Similarly, Mitra et al. [21] performed viral metagenomic sequencing on DNS obtained from feedlot steers with acute BRD (n = 47) and healthy/asymptomatic feedlot steers (n = 46) on six farms in Mexico and four in the U.S.A. In this study, 21 different viruses were identified. However, in the statistical analysis, IDV was the only virus moderately associated with BRD (odds ratio (OR)− = 2.94). Furthermore, showing an OR > 1, the presence of IDV can be considered as a moderately important risk factor for BRD. Likewise, the amount of IDV RNA in samples from BRD animals was also significantly higher than that from asymptomatic IDV-positive animals.

The detection of the viral agent does not necessarily imply that it is the cause or is related to BRD. The upper respiratory tract, where the nasopharynx is located, is an entry point for many pathogens that cause respiratory infections. The nasopharynx is not sterile, so the existence of resident or transient bacterial microbiota, as well as viruses, is normal—if the microorganisms present in this microbiota are potential pathogens, they are sometimes referred to as pathobionts [22]. However, as long as the defense mechanisms of cattle are not altered, this microbial population does not proliferate or colonize the lower respiratory tract, causing pathology. The lower respiratory tract (trachea, bronchi, bronchioles, and alveoli) is considered sterile, that is, the presence of some pathogenic microorganism can be associated as being responsible for BRD since it is an area in which there should be no presence of microorganisms. Thus, the presence of IDV in the nasopharynx may be causative of BRD, or part of the pathobionts, without causing any pathology. This theory is reinforced by the fact that IDV-positive animals have been detected by RT-PCR in DNS samples from healthy and BRD-asymptomatic animals (Table 2).

In some of the studies where the upper respiratory tract was sampled, more tests would have been needed to confirm whether IDV was involved in BRD. IDV was detected more frequently in calves with BRD, than in healthy ones. Studies with samples collected from calves with clinical signs of respiratory disease suggest a positive relationship between BRD and IDV infection [4], although the actual involvement of IDV in BRD severity in the field remains to be demonstrated [3].

### 3.2. Epidemiology

Serological antibody tests suggest that IDV circulates in domestic animals, including cattle, pigs [30,32,33], feral pigs [9], wild ruminants [44], camelids [42] horses [45,46], and small ruminants (sheep and goats) [28,33,47], in addition to being potential zoonosis for humans [8,48,49]. Cattle are the natural reservoir (main reservoir) of the virus, as indicated by the widespread and high antibody titers against IDV and its common isolation in cattle. Of special interest is the prevalence of 94.6% in sera from cattle at slaughter, in Ireland [33]. In contrast, antibodies against IDV have not been detected in poultry, although recent molecular diagnostic studies have revealed the presence of the IDV genome in poultry farms in Southeast Asia. In this study, it was not possible to carry out a complete sequencing of the genome and identify the lineage of the virus. Nevertheless, these findings suggest adding this species to the growing list of hosts susceptible to IDV [50].

Prevalence studies have shown a worldwide distribution of IDV in bovine species (Table 2), such as in Denmark [51], France [28], Italy [3], Ireland [19], Luxembourg [32] and the United Kingdom [34], U.S.A. [23], Mexico [21], Argentina [27], China [36], Japan [39], Turkey [41], Benin, Morocco, and Togo [42].

The transmission routes of IDV can be through direct contact with infected individuals or via aerosol at short distances. A third route of IDV transmission, through contaminated fomites, is suspected but has not yet been confirmed. The direct contact transmission route was demonstrated by experimentally inoculating a group of cattle with IDV and, twenty-four hours later, introducing three seronegative calves to this same virus (exposed calves) to this same group; the three exposed calves seroconverted between 9 and 13 days post-challenge in the hemagglutination inhibition (HI) assay [52]. Viral excretion, determined by quantitative RT-PCR on nasal swabs, showed that exposed calves that were introduced with cattle that had been inoculated with IDV shed virus for 6 to 9 days post-challenge [52]. To demonstrate via aerosol transmission, Salem et al. [4] carried out a study where three calves (sentinel calves) were housed three meters apart, with steel panels separating them from calves that had previously been inoculated with IDV (without contact, but subjected to aerosol infection). In the group of sentinel calves, one out of three calves was infected with IDV via aerosol (positive to RT-PCR in nasopharyngeal swabs). The excretion of the virus in this calf began 11 days after the inoculation of IDV in the group of directly inoculated calves. This sentinel calf then infected the rest of the calves in its pen through direct contact. Furthermore, IDV was detected in air samples collected from different areas: first in the pen of calves inoculated with IDV, then in the intermediate zone of separation of the two pens, and finally in the sentinel calves’ pen. These findings demonstrated that IDV can be airborne transmitted and may infect calves over short distances. To date, the minimal infective dose of IDV in cattle is unknown, although studies suggest that the transmission rate of IDV is high in cattle through natural transmission mechanisms (direct contact and aerosol) [4,52], as was also indicated by epidemiological data showing a worldwide distribution of this pathogen.

The studies suggest that cattle trade could play a role in the observed differences in IDV seroprevalence worldwide. In fact, the overall prevalence in cattle in EU importing countries (i.e., Italy with a mean seroprevalence of 92.4%) is higher than in exporting countries (i.e., France with a mean seroprevalence of 47.2%) [3,28]. This leads to the hypothesis that transportation may be one of the causes of viral spread of IDV among young calves in importing countries. The spread of IDV and its role in the occurrence of BRD after transport is still unclear, and further studies should track IDV in calves traded between countries or regions. However, post-transport pathogen shedding has been shown to increase due to immunosuppression of young calves, not only for bacteria such as *M. haemolytica*, *M. bovis*, and *P. multocida*, but also for viruses such as BCoV and BRSV [53]. Although it is clear that transport can be a risk factor for the transmission of IDV, it should be noted that the type of production system to which the calves are subject should also be taken into account when comparing the different seroprevalences between countries or regions. Nevertheless, to decipher the real impact that transport has on the development of IDV, two groups of calves with the same health status and origin should be studied in each country and one group could be transported a short distance and the second group a longer distance.

An epidemiological study carried out in Mississippi (U.S.A.) sampled 137 recently weaned grazing calves with nasal swabs, which were analyzed by RT-PCR. Calves were from different origins and submitted to a long transport before arriving to the fattening farm. The proportion of IDV-positive nasal swabs was higher in calves with respiratory disease (29.1%) than in healthy/BRD-asymptomatic calves (2.4%) [23]. The same authors observed passively acquired maternal antibodies in 448 serum samples analyzed by HI assay from newborn calves over a two-year period, reporting an IDV seroprevalence of 94%. This would show that most of the dams would have been, at least, in contact with the IDV. In addition, 484 other calves aged 6–8 months old were sampled, reporting a seroprevalence of 5.8%. This demonstrates that after 6 months of age, the maternal immunity is drastically reduced and many calves can be vulnerable to IDV infection. Finally, it was also demonstrated that seropositivity to IDV increased from the first year of life onwards, with values higher than 54.2% from ages 1 to 14. It has also been reported, by two serological studies that analyzed stored sera, that IDV was already prevalent in cattle in Mississippi (U.S.A.) as far back as 2004, and in Nebraska (U.S.A.) since 2003 [10,23]; the overall seroprevalence rate determined by HI assay was 15.9% and 81.9%, respectively. The highest seroprevalence rate (81.9%) was observed in cattle aged 2 years or older, while the lowest seroprevalence rate (15.9%) was observed in samples collected from a single farm from calves between 6 and 8 months, and cows over 1 year of age (Table 2).

The lag of time existing in the research works, between the evidence of circulation (2003) and its correlation with BRD (2014), can be attributed to the fact that when the virus first appeared, it caused a very mild disease, with hardly any clinical signs or lesions, and, therefore, went unnoticed in the first moments of its circulation in bovine populations. However, as it has been replicating in calves and transmitting between them, mutations and new variants might have occurred. Additionally, being well adapted to the host may have increased IDV virulence and replication efficiency [11,52]. The presence of these more efficient IDV variants could be one of the reasons why BRD incidence has increased these recent years [7].

The presence of IDV in South America has also been demonstrated by serological studies. In a study from Argentina, 73% of the farms analyzed had at least one positive animal. Of the 165 serum samples from bulls over three years of age that had been collected in 2013, originally to estimate the seroprevalence of reproductive diseases by HI assay, 68% were seropositive to IDV (Table 2) [53]. Molecular detection of IDV from a case of bovine respiratory disease has also been reported in Brazil, highlighting the importance of investigating IDV as a possible causative agent of respiratory disease [16].

The presence of IDV in Europe was first reported in France in 2015, but had been circulating since 2011. After this initial finding, IDV was found to be in other European countries: Italy, Luxembourg, Ireland, and the United Kingdom. The prevalence of IDV in Europe was similar to that found in North America, where the virus and/or antibodies were detected in multiple animal species, with the highest prevalence found in cattle (Table 2) [11].

In France, samples (lung fragments, DNS, and transtracheal aspiration fluids) were taken from 134 calves (healthy/asymptomatic for BRD and clinically sick with BRD) between 2011–2014. Samples were analyzed by RT-PCR, and six samples were positive for IDV (4.5%), where five positive samples were lung tissue and one sample was a deep nasal swab (Table 2). Coinfections with *P. multocida*, *M. haemolytica*, *H. somni*, BRSV and/or BoHV-1 were detected in four out of the six IDV positive samples. In the other two samples, no coinfections with the respiratory pathogens analyzed were detected, despite both animals suffering from clinical signs; nevertheless, these two animals were treated with antibiotics, which could have interfered with the diagnosis [29].

Additionally, in France [28], a serological study was performed on bovine sera (n = 3326) collected from 2014 to 2018, in five regions. Sera were analyzed by HI assay. All animals were older than one year of age, excluding interference with antibodies of maternal origin. The resulting global seroprevalence was 47.2%, but the results varied depending on the geographical region (31.0–70.0%) (Table 2). In Ireland, during 2014–2016, 320 bovine nasal swab samples from 84 farms were tested for BRD by RT-PCR. It was determined that 18 calves (5.6%) were positive for IDV from 10 different farms (11.9%). Five of the IDV positive calves (27.8%) were not positive for any other viral pathogens under study (BoHV-1, PI3V, BCoV, BRSV and BVDV). This finding does not unequivocally demonstrate that the IDV was solely responsible for the BRD (Table 2) [19]. Additionally, in Ireland, in 2017, sera samples were collected from cattle in slaughterhouses across the country (n = 1219) and tested for IDV antibodies, as well as bovine sera (n = 1183), that had been collected for BRD diagnostics between 2016 and 2017, also tested for antibodies against IDV. A large difference was observed, in terms of seroprevalence, between the two studies, with 94.6% in sera from bovines at slaughter and 64.9% in sera from animals pre-sampled for BRD diagnosis (Table 2) [33]. The presence of IDV has been reported in Italy, as well. In an epidemiological study executed in 574 farms to detect IDV by RT-PCR between the years 2014 and 2016, a total of 895 samples of nasal swabs and lung tissues were analyzed. These samples came from cattle with BRD (n = 603) and without BRD (292 samples of dead animals sent to the laboratory and diagnosed with diseases other than BRD). Of the samples taken from cattle with BRD, there was a prevalence of 8.0%, and of the samples taken from cattle without BRD, there was a prevalence of 3.4%. Of the 48 IDV positive samples that came from cattle with BRD, in 62.5% of the cases, IDV was the only viral agent detected among those included in the laboratory analysis (BoHV-1, PI3V, BCoV, BRSV, and BVDV). This further supports the hypothesis that IDV may play a primary role in the occurrence of BRD, though this result can be confounded by the limitations of the diagnostic techniques used for the detection of other BRD pathogens. In 37.5% of the remaining samples, IDV was found together with other respiratory viruses, especially BCoV (Table 2) [3]. Another finding was that samples from cattle with BRD that were IDV positive, were taken from the lower respiratory tract (lung tissue), although in a smaller proportion (3.4%) than those that came from the upper respiratory tract (nasal swab) (9.4%). This reinforces the finding from experimental infection studies that the upper respiratory tract is probably the preferred site of replication for this virus [3]. Additionally, in Italy in 2015, a serological study performed with HI tested 420 sera samples from 42 dairy farms, with a prevalence of 92.4% (398 positive samples) of IDV detected in 100% of the farms. The results obtained in this epidemiological study demonstrated the active circulation of IDV in Italian beef and dairy cattle farms and strengthen the hypothesis of the association of BRD and IDV, which may play an important role in the appearance of this disease respiratory pathology [3]. In Luxembourg, a high IDV seroprevalence (80.2%) was also observed in bovine sera from cattle with ages between 23 and 209 months (n = 450) collected during 2012–2016 (Table 2) [32].

In Asia, IDV was reported for the first time in 2014, specifically in China. It is assumed that IDV has been circulating in Asian cattle since 2010. Although some studies have shown the circulation of IDV in Asian countries, more data are needed to better understand the epidemiology of IDV in cattle in this continent [11,15]. In Japan, a recent study on the seroprevalence of IDV found values ranging between 45% and 71%, with a mean of 57% in the 960 sera collected from cattle over 24 months of age between 2009 and 2018, and in 96 different farms, demonstrating the circulation of the virus in Japan for at least 10 years [40].

In Africa, IDV has been known to circulate in cattle since 2012. IDV RNA has been detected in bovines, giraffes, and wildebeest from Namibia. It also seems that the African IDV sequence was distinct from any other sequence for all its seven segments and most likely represents an African-specific genotype within the D/OK lineage [18]. The infection seems less widespread in calves from African countries than in Europe or America. This could possibly be due to lower animal density in the livestock industry, as overcrowding has been seen to contribute to further spread of IDV infection [53].

All the results reporting prevalence data obtained in this review are summarized in Table 2. It is important to remark that comparisons between different studies should be carried out carefully, since there might be variability depending on the time of sampling, the analytical method, time after the IDV exposure, and the animals’ age.

### 3.3. Pathogenesis

Cattle are the main reservoir of IDV, with the virus having been detected in the following tissues of infected individuals: nasal cavity, trachea, bronchioles, and lung lobes (cranial, middle, accessory, and caudal lobes) at 8 days post-inoculation. It has also been detected in the tracheobronchial and mediastinal lymph nodes [4].

The highest viral load of IDV was observed in the nasal cavity, specifically in the ethmoid nasal concha. High IDV RNA loads were also found in the olfactory bulb and tonsils in sentinel animals that were infected via aerosol, but IDV tropism for these tissues could not be confirmed by IHC or virus isolation, and further studies are needed to confirm this finding [47]. IDV has tropism for the upper respiratory tract and replicates preferentially in the epithelial cells of this anatomical region. It also replicates in the lower respiratory tract, being able to induce mild or moderate interstitial/bronchointerstitial pneumonia [4]. It is unknown in which cells IDV replicates in the lung: the chronology of how it replicates and the mechanism by which the virus causes damage to the respiratory system after the animal is infected, remain unclear.

IDV shows optimal growth at both 33 °C and 37 °C in cell cultures, suggesting that elevated temperature does not limit IDV replication in the lower respiratory tract, where temperatures are higher than in the higher respiratory tract [54].

Using the ELISA technique, seroconversion to IDV specific IgG was detected in all directly inoculated animals 10 days after exposure. In addition to the humoral immune response, IDV also induces a cellular immune response. The mean excretion length of IDV is 8.1 ± 1.9 days [4].

The IDV genome has been detected in serum samples from seriously sick cattle, which implies that the virus could temporarily enter the circulatory system of animals (viremia) and spread to other organs [11]. IDV has been detected by RT-PCR in feces on day five post infection and in the jejunum on day six post infection, which corresponds to the time of greatest viral replication in the respiratory tract. Although intestinal tropism and infectivity must be confirmed by IHC and/or isolation studies, these results suggest the possible IDV fecal excretion route through the digestive tract, in addition to the oronasal excretion already demonstrated. It is also suggested by Yu et al. [11] that IDV could replicate within the intestinal tract in a similar way to IAV and IBV. This possible enteric tropism of IDV could be due to the high acid stability of this virus. In addition, the high thermal and acid stability of the virus means that IDV has a high resistance potential abroad, which could explain its high transmission efficiency [4].

A study was conducted to determine the coinfection effects of IDV and *M. haemolytica*. Calves were inoculated intranasally on day 0, with an IDV strain (D/Bovine/C00046 N/Mississippi/2014), and again on day 5 intratracheally with the *M. haemolytica* D153 strain (serotype A1), the main pathogen causing BRD. Surveillance data suggest that *M. haemolytica* and IDV coinfection occurs in calves, although primary IDV infection has not been shown to potentiate or aggravate lung pathology or clinical signs [6]. As a limitation of this study, it was proposed that they were healthy individuals housed in controlled experimental environments, and not in real field conditions. Additionally, it was a sequential coinfection (separate infection in time), not a simultaneous one. Thus, an earlier inoculation of the bacteria could have been more optimal to obtain a more severe clinical outcome. Moreover, the inoculation of *M. haemolytica* was into the mid-trachea and avoided a significant portion of the upper respiratory tract affected by IDV, which may have reduced the opportunity for synergism between the two respiratory pathogens. In any case, it is still pending to demonstrate unequivocally that a simultaneous coinfection of *M. haemolytica* and IDV could increase the synergy in clinical signs and virulence of BRD due to optimization of dose and route of inoculation [11,55].

The effects of the simultaneous coinfection of IDV and *M. bovis* in cattle were also studied. Two-month-old calves were infected via aerosol and simultaneously, with a French IDV strain (D/bovine/France/5920/2014) and with the *M. bovis* RM16 strain [55]. The study included four experimental groups, with five calves per group: control, IDV inoculated, *M. bovis* inoculated, and IDV + *M. bovis* inoculated. IDV was shown to facilitate the replication of *M. bovis* by modulating the local innate immune response. This coinfection generates a strong immune response in the lower respiratory tract, especially of interferon gamma (IFN-γ), by increasing the expression of IFN-γ gene. This gene was found to be overexpressed after coinfection, positively correlating with disease severity, immune response, and white blood cell recruitment in the lungs. Calves inoculated with IDV began to show clinical signs at 4.4 ± 1.1 days, reaching the onset of clinical signs at 7.6 ± 1.8 days, with signs of disease during 8.0 ± 1.2 days. Calves infected with IDV showed mild to moderate respiratory clinical signs. These signs included: nasal discharge, cough, slight increase in respiratory rate (35 to 40 breaths/minute), and mild dyspnea with abnormal lung sounds, for the most affected calves, without affecting appetite, body temperature, or general condition. Calves that were inoculated with *M. bovis* began to show clinical signs later, at 7.8 ± 1.0 days, reaching the onset of clinical signs at 13.4 ± 1.5 days and with a duration of clinical signs of 9.6 ± 1.9 days. Clinical signs suggested an infection of the upper respiratory tract, trachea, and bronchi. They were mainly characterized by a strong and frequent cough, fever, tachypnea and, to a lesser extent, mucus secretion and mild dyspnea. Calves coinfected with IDV and *M. bovis* developed clinical signs similar to those observed in calves infected with *M. bovis* alone, except that the clinical signs were more severe and began earlier (5.0 ± 1.6 days), with a maximum onset of clinical signs of 8.6 ± 1.3 days and duration of clinical signs of 9.8 ± 2.5 days. Two out of five calves showed significant clinical signs: hypo-/anorexia, poor general status, increased respiratory rate, mucopurulent nasal discharge, repeated spontaneous cough and abnormal lung sounds. The other three calves showed similar clinical signs but less severe. All animals recovered 21 days post inoculation except one, which was still slightly ill. These data suggest that IDV facilitates the pathology caused by *M. bovis*. Calves coinfected with IDV and *M. bovis* showed statistical differences in the mean clinical score of each group per day compared with calves infected only with IDV at 4, 5, 8 and 9 days post infection. In addition, statistical differences were observed between 3 and 8 days post infection between calves coinfected with IDV and *M. bovis* and those infected only with *M. bovis*. Macro and microscopic lesions were more severe in coinfected calves than in those infected with IDV or *M. bovis*. The main macroscopic lesions observed in the coinfected calves at six days post infection were severe tracheitis with necrosis, fibrin purulent exudates on the mucosal surface, and interstitial pneumonia with atelectasis in the cranial lobes, right middle lobe, and accessory lobe. At 21 days post infection, four of the five remaining coinfected calves presented an acute interstitial pneumonia of minimal extension (5 to 10% of the lung parenchyma). Calves infected with IDV presented microscopic lesions in the nasal cavities and/or in the trachea, characterized by the loss of cilia, necrosis and erosion of the superficial epithelium of the mucosa, and infiltration of the lamina propria by mononuclear cells. Coinfected calves with IDV and *M. bovis* euthanized at 6 days post infection, showed rhinitis and tracheitis lesions similar to those of the IDV-infected calf group, except that these lesions were more pronounced in the trachea. Only one coinfected calf presented mild microscopic lesions of subacute bronchointerstitial pneumonia in the left cranial lung lobe, characterized by the presence of neutrophils in the bronchial lumen, neutrophilic and macrophagic alveolitis, and peribronchial and septal lymphoplasmacytic infiltration in the lung. Coinfection with IDV and *M. bovis* also increased recruitment of white blood cells to the airway lumen, but differences were not significant. IDV infection promotes colonization of *M. bovis* in both upper and lower respiratory tract. Bacterial titers (DNA copies) of *M. bovis* were evaluated by quantitative PCR on nasal swabs collected between 2 and 20 days post infection. The replication kinetics of *M. bovis* in this period showed higher values of DNA copies in calves coinfected with IDV, finding a significant increase at four days post infection. Positive samples of bronchoalveolar lavage fluid to *M. bovis* also showed higher bacterial titers (number of *M. bovis* DNA copies) in calves coinfected with IDV at seven days post infection.

All IDV infections—single or coinfections—induce a humoral response in cattle. In the study carried out by Lion et al. [55], all calves were seronegative for IDV and *M. bovis* before the infections. To evaluate the humoral response, antibodies against IDV were determined in the sera by HI assay. Calves seroconverted against IDV from seven days post-infection, both in the group infected with IDV and with IDV + *M. bovis*. Immunoglobulin A (IgA) and IgG specific for IDV were detected by ELISA technique in the sera of calves infected with IDV or IDV + *M. bovis* after seven and ten days post infection for IgA and IgG, respectively. Antibodies against *M. bovis* were not detected by ELISA in any sample, suggesting a delayed humoral response against this bacterial infection. These results demonstrate that IDV infection induces a fast local (IgA) and systemic (IgG) humoral immune response.

In summary, the IDV and *M. bovis* coinfection shortened the incubation period, worsened the clinical signs of BRD, and caused more severe macroscopic and microscopic lesions, including more signs of pneumonia, compared with calves infected with a single pathogen. Furthermore, IDV promoted upper and lower respiratory tract colonization by *M. bovis*, and increased white blood cell recruitment [55]. Further studies with a larger population exposed under field conditions are needed to understand the possible synergistic effects of IDV with other pathogens, especially those involved in lung pathology [11].

### 3.4. Clinical Signs

The studies based on experimental infection with IDV of seronegative calves showed that it caused respiratory signs, ranging from mild to moderate, associated with inflammation and damage at sites of viral replication (upper respiratory tract epithelial cells and in the lower respiratory tract) [4,52]. Most observed clinical signs were: spontaneous to repeated dry cough, unilateral or bilateral serous/mucoid nasal discharge, serous eye discharge, depression, mild to moderate tachypnea (35–65 breaths/minute) and in severe cases, dyspnea and abnormal lung sounds (wheezing). It has not been reported to cause hypo-/anorexia or fever.

Clinical signs of IDV began at 4.6 days ± 1.5 days, with a maximum of clinical signs at day 8.0 ± 0.8. The duration of clinical signs was 6.4 days ± 2.5 days. All calves in the study recovered after 12 days of IDV inoculation. The possible presence of asymptomatic carrier individuals has also been described [52]. These clinical signs may vary due to the possibility of IDV coinfecting with other respiratory pathogens and causing BRD. Under field conditions, with uncontrolled risk factors such as adverse weather conditions, transportation, mixing of animals or high density, the severity of the disease induced by IDV compared with that observed in the experimental challenges could be greater [4,52].

Another factor to consider in the variability of the BRD severity is the routes of infection and the dose. The calves infected in the experimental challenge by direct contact presented slightly more severe clinical signs than the animals that were directly inoculated with the virus. This highlights that the virus may have acquired host adaptations during transmission, which could have facilitated more efficient replication. Mutations due to adaptation to the host in influenza viruses have been demonstrated [52].

### 3.5. Lesions

Microscopic lesions of IDV infection were determined by histological analysis of sections of the following tissues stained with hematoxylin and eosin (H/E): nasal cavity (nasal turbinates and nasal mucosa), trachea, bronchi, bronchioles, and lung. The following lesions associated with the infection of this virus have been found, although all these lesions do not have to occur simultaneously:

Rhinitis: Inflammation of the nasal turbinates and nasal mucosa with increased neutrophils and infiltration of the lamina propria by mononuclear cells in the nasal epithelium. In some cases, the superficial nasal epithelium of the mucosa showed mild multifocal loss of cilia, necrosis, and erosion. Suppurative rhinitis with increased neutrophils mixed with mucoid secretion in the nasal epithelium has been described in some calves [7,52,55].

Tracheitis: Moderate inflammation of the trachea was observed with a significant increase in neutrophils in the mucosa and submucosa, multifocal areas of epithelial infiltration of neutrophils, an occasional loss of cilia, and thickening of the epithelium. In some cases, the superficial epithelium of the mucosa suffered necrosis and erosion [7,52,55].

Bronchitis and Bronchiolitis: An increase in neutrophils was observed in the bronchial lumen [52].

Interstitial/bronchointerstitial pneumonia: Increased neutrophils in the bronchial lumen, neutrophilic and macrophagic alveolitis, and peribronchial and septal lymphoplasmacytic infiltration in the lung [4,55].

It was concluded that IDV causes a milder respiratory tract infection compared with other viruses (i.e., BRSV) [7]. The clinical impact of the IDV in this study was moderate, as calves recovered quickly from the infection, indicating a milder pathogenicity in calves’ lower respiratory tract compared with other respiratory viruses. The milder pathogenicity may also be due to the conditions of the experiment (these were not field conditions), the strain of IDV used, the dose and route of inoculation (D/bovine/Kansas/162655/2012 strain, 6.0 TCID_50_/mL in MEM, intranasal), and/or the host (age of the animal and state of the immune system). The factors described above may also affect differences in macroscopic and microscopic lesions observed in IDV pathogenesis studies [4].

Regarding the macroscopic lesions, irregular areas of atelectasis (collapsed lung parenchyma) with an intense red coloration were observed. The lesions were limited to the cranial lobes, covering between 5% and 10% of the entire lung surface. No macroscopic lesions were found in the nasal cavity, larynx, or trachea [4].

### 3.6. Diagnosis

IDV can be detected in both the upper and lower respiratory tract. The samples taken for molecular, pathological, and microbiological diagnosis are based fundamentally on deep nasal (nasopharyngeal) swabs/nasal swabs, where a significantly higher positivity rate is found, but also through fluids from bronchoalveolar and transtracheal lavage. Sampling lung tissue, trachea, and nasal turbinates during necropsy is also useful [3,29,55]. Blood serum is the sample used for serological diagnosis [4].

Several analytical techniques have been described as possible for the diagnosis of IDV. These diagnostic techniques cover both the detection of the pathogen in the animal and the contact with the virus. RT-PCR is used for molecular diagnosis (detection of the causal agent), with high sensitivity and specificity [12,56]. It is important to point out that most real-time RT-PCRs are aimed at detecting the IDV PB1 gene, which is one of the most conserved and stable regions within the seven RNA segments of the IDV genome. The inclusion of IDV diagnosis in multiplex PCR for BRD could be recommended, and veterinarians would have more information on BRD-causative pathogens in the herd. HI and ELISA assays are used for serological diagnosis (detection of antibodies) [12]. Histology with H/E staining can be used to detect the microscopic lesions caused by the IDV, and immunohistochemistry (IHC) can demonstrate the presence of the virus in the analyzed tissue [4].

### 3.7. Control and Prevention

Cattle play an important role in the replication and spread of IDV, highlighting the need for further surveillance and risk assessment of this emerging virus in cattle [12]. Avoiding and controlling the classic risk factors to prevent BRD should be a priority to avoid cases of IDV infection. Some of the best practices include isolation for at least four weeks for all purchased calves arriving at a farm, but also regular cleaning and disinfection of equipment, proper storage of feed and water, as well as preventive measures such as vaccination for some of the BRD pathogens and biosecurity measures that prevent the presence of IDV carriers [12,53].

There are currently no vaccines for IDV in animals, probably due to its recent discovery [12]. As with other types of influenza viruses, IDV will constantly evolve into more diverse lineages or strains [7].

Studies have shown the protective efficacy of a DNA vaccine expressing the HEF glycoprotein against the two IDV lineages (D/OK and D/660) in guinea pigs. It was shown that the vaccinated animals showed significant titers of neutralizing antibodies against both IDV lineages and, when performing the intranasal challenge with both IDV lineages, the guinea pigs were protected, since no viral RNA by quantitative RT-PCR was detected in the necropsied respiratory tissues of the vaccinated animals. Although further studies are needed to assess its protective efficacy against other IDV lineages in guinea pigs and to determine whether protection against various IDV lineages can be observed in cattle, these results suggest that a DNA vaccine expressing the glycoprotein HEF has the potential to protect animals from IDV infections [57].

In another experimental study, an inactivated IDV vaccine in calves was tested. It was found to be immunogenic and provided partial protection against IDV. The inclusion of IDV in commercial BRD vaccines could improve their effectiveness in its prevention.

There is no vaccine or specific treatment for IDV, and it is considered that it would be very interesting to include this new viral pathogen in existing commercial bovine vaccines, generating herd immunity against IDV, since it is a pathogen that is widespread on farms worldwide and associated with the occurrence of BRD. If an effort was carried out to include IDV in commercial vaccines, animal health, animal welfare, and productivity would be improved, and treatment costs and use of antibiotics would likely decrease.

Although it is unknown whether IDV infection in humans causes disease, vaccination in cattle could help limit zoonosis [7]. The probability of transmission of this pathogen would be reduced between cattle, but also between cattle and humans, since IDV is a potential zoonotic pathogen. Although its importance for public health seems be low, human health should not be forgotten in a One Health framework.

### 3.8. Importance for Human Health

Successful zoonotic infections occur when pathogens acquire the ability to cross the species barrier and replicate effectively in the new host (humans). Findings to date suggest that IDV infects multiple species and has zoonotic potential to infect humans, but further studies are needed to understand its complexity and evolution [11]. The IDV that was first identified in 2011 shared approximately 50% homology with human ICV, and the isolation of IDV from both pigs and cattle indicates the possibility and potential of IDV for successful transmission to other mammals [12].

Although studies have reported that IDV has not shown a drastic antigenic shift over the years, the unpredictability of influenza viruses and their ability to frequently mutate makes them a potential threat to human health. Second to IAVs, IDV is the influenza virus with the highest capacity to infect a wide range of hosts, hence its zoonotic potential and global concern. The ability of IDV to cause disease in humans has not yet been thoroughly investigated, and it is not clear whether this virus can sustain human-to-human transmission, nor is there evidence that it causes disease in people, but it has been demonstrated that this virus can replicate and be transmitted by direct contact in ferrets and guinea pigs (surrogate models of human influenza infection). Recent studies have confirmed that IDV is able to replicate efficiently in well-differentiated human airway epithelial cells (HAECs), which is an in vitro model of the human respiratory epithelium used to assess the zoonotic potential of emerging respiratory viruses [58]. However, very few mutations have been observed in circulating IDVs, suggesting that the evolution of the virus is slow. In addition, the HEF is stable and the main target of neutralizing antibodies generated during IDV infection, indicating that mutations escaping from antibodies are rare.

A serological study conducted in the U.S.A. demonstrated the existence of specific antibodies against IDV present in people, finding a prevalence of 1.3% in 316 samples of the human population during 2007–2009 [8]. IDV may pose a potential threat as an emerging pathogen to personnel who are in contact with cattle. A very high seroprevalence of 91.0% was observed in Florida (U.S.A.), in samples analyzed by HI, from participants who had reported a minimum of weekly contact with cattle during the six months prior to sampling (n = 35) [49]. In Europe, it was reported that antibodies against IDV were present and have increased over time in the Italian population between 2005 and 2017 (n = 1281), obtaining a mean value of 26.2% through HI, and reaching maximum seroprevalence values in 2014 (46.0%). The prevalence of antibodies against IDV in humans implies that the virus can infect humans and pose a potential threat to human health [59].

In 2017, a study showed that the IDV genome was detected by RT-PCR in a nasal wash sample from a worker at a pig farm in Asia. In addition, molecular surveillance of known respiratory viruses, including IDV, with bio-aerosol sampling (ambient air sampling) at an international airport in North Carolina (U.S.A.) in 2018 detected, by RT-PCR, that in four out of twenty-four samples positive for respiratory pathogens, one was specifically positive for IDV, noting that none of the twenty-four samples were positive for IAV, IBV, and ICV [60]. In addition, in 2017–2018, the IDV genome was detected by RT-PCR in bio-aerosol samples in a hospital emergency room in North Carolina [61]. All these studies demonstrate that IDV could be capable of infecting and spreading between humans. Overall, humans do not have pre-existing immunity against this new influenza virus [62].

In the context of One Health, animal health is an important link. The consumption of animal products has been a rapidly growing component of the food industry in recent decades. Therefore, continuous surveillance of emerging pathogens in production animals is needed to ensure animal welfare (animal health), and also to prevent zoonosis (human health) [53].

To sum up, despite the zoonotic potential of IDV, as long as the pathogenic potential of the virus does not change drastically (mutation) its importance for public health seems to be low. However, since the features and changes of influenza viruses are difficult to predict, there is a need to monitor the virus more closely, especially in humans, and ideally, make an effort to design a prototype vaccine that would protect humans in case of an epidemic outbreak [54].

## 4. Conclusions

IDV is a new pathogen that was first identified in 2011, although it has been circulating in cattle populations since at least 2003. It is a virus that is highly resistant to pH and temperature, and has broad cell tropism and number of hosts, in which cattle are the main reservoir where IDV has a worldwide distribution.

IDV may play a role in BRD, being capable of causing mild to moderate respiratory pathology on its own, with a high rate of transmission, and can predispose or enhance the effects of other respiratory pathogens, such as *M. bovis*, where coinfection worsens the clinical signs and severity of the disease.

Transmission of IDV is by direct contact and by aerosol over short distances. IDV has a tropism for the upper respiratory tract, but it also replicates in the lower respiratory tract and can cause interstitial/bronchointerstitial pneumonia.

The clinical signs of IDV range from mild to moderate, and are associated with inflammation and lesions at sites of viral replication.

Most newborn calves acquire antibodies against IDV through the colostrum, which remain high until 3–4 months old, after which they decrease and cattle are most vulnerable to IDV infection.

There are no vaccines or specific treatment for IDV. However, the efficacy of some experimental vaccines in cattle has been tested and has shown encouraging results in controlling the disease.

Although IDV public health importance appears to be low, it has zoonotic potential and there are already reports of high IDV seroprevalence in cattle farm personnel.

Further research and monitoring of this new virus is needed to better understand its role in BRD, its zoonotic potential, and the development of effective vaccines and treatments.

## Figures and Tables

**Figure 1 viruses-14-02717-f001:**
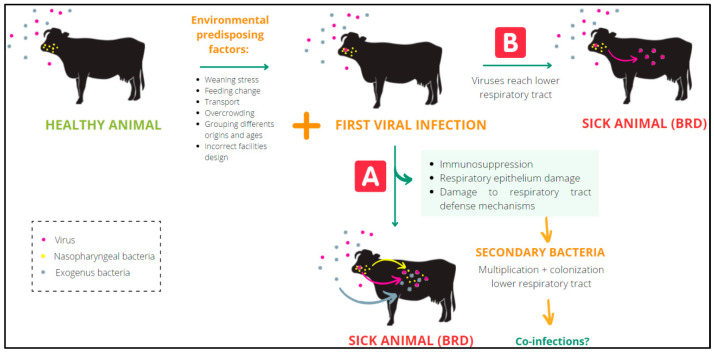
Mechanism of viruses predisposing the infection of bacteria in BRD.

**Figure 2 viruses-14-02717-f002:**
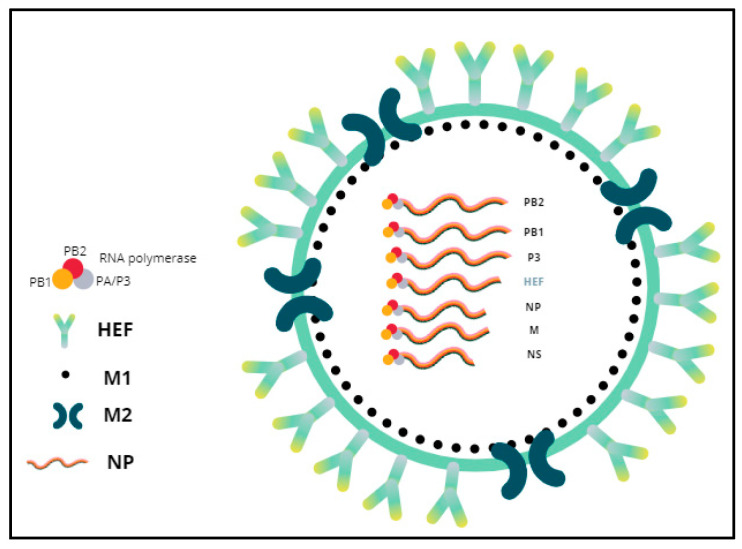
General morphology of influenza D virus. The envelope has membrane proteins HEF and M2. The M1 protein is underneath the membrane surface and encases the virion core that contains the genome. Each vRNA segment forms a vRNP together with the NP and RNA polymerase complex (PB2, PB1 and PA/P3).

**Table 1 viruses-14-02717-t001:** Detail of the data search and key words used in Google Scholar, PubMed, Scopus and Web of Science (accessed from November 2021 to October 2022). For important topics, full words and their most commonly used abbreviations in the scientific literature were used with the Boolean operators AND, OR, and NOT.

Category	Group	Key Words
General	Bovine respiratory disease Influenza D virus	Influenza D Virus, IDV ^1^, Bovine, Cattle, Calves, BRD ^2^, Bovine Respiratory Disease, Bovine Respiratory Syndrome, BRS ^3^
Specific	Etiology	Influenza D Virus, Etiology, Origin, Bovine, Cattle, Calves
	Epidemiology	Influenza D Virus, Epidemiology, Prevalence, Seroprevalence, Bovine, Cattle, Calves
	Pathogeny	Influenza D Virus, Pathogenesis, Bovine, Cattle, Calves
	Clinical signs	Influenza D Virus, Clinical signs, Bovine, Cattle, Calves
	Lesions	Influenza D Virus, Bovine, Lesions, Bovine, Cattle, Calves
	Diagnostics	Influenza D Virus, Diagnostic, Diagnosis, Bovine, Cattle, Calves
	Treatment	Influenza D Virus, Treatment, Bovine, Cattle, Calves
	Control and prevention	Influenza D Virus, Prevention, Preventive measures, Treatment, Vaccine, Bovine, Cattle, Calves
	Public health	Influenza D Virus, Zoonotic, Zoonosis, Public Health, Bovine, Cattle, Calves

^1^ IDV: influenza D virus. ^2^ BRD: bovine respiratory disease. ^3^ BRS: bovine respiratory syndrome.

**Table 2 viruses-14-02717-t002:** Summary of IDV prevalence (%) by country, with the age of the animals expressed as >6 months old, =6 months old or <6 months old, year of the sampling, analytical method, and references. Sample size is reported in brackets. Prevalence is reported as total if the study did not detail if the animals were showing clinical signs compatible with BRD. Prevalence is reported as BRD if the study was on animals with clinical signs and as healthy/asymptomatic if the study reported the results on animals without BRD clinical signs.

Country	Reference	Year Sampling	Analytical Method		Prevalence (N)		
				Age of Animals in Months	BRD ^1^	Healthy/Asymptomatic	Total
U.S.A.	Luo et al. [10]	2003–2004	IH ^2^	>6	-	-	81.9% (293)
U.S.A.	Ferguson et al. [23]	2004–2006	IH ^2^	>6	-	-	15.9% (605)
U.S.A.	Ferguson et al. [23]	2013–2014	IH ^2^	<6	-	94.0% (448)	-
U.S.A.	Silveira et al. [24]	2014–2015	IH ^2^	>6	-	-	77.5% (1992)
U.S.A.	Hause et al. [25]	2013	RT-PCR ^3^	ND ^4^	-	-	18.0% (45)
U.S.A.	Collin et al. [26]	2014	RT-PCR ^3^	ND ^4^	4.8% (208)	-	-
U.S.A.	Ferguson et al. [23]	2014	RT-PCR ^3^	>6	29.1% (55)	2.4% (82)	
Argentina	Álvarez et al. [27]	2013	IH ^2^	>6	-	-	68.0% (165)
France	Oliva et al. [28]	2014–2018	IH ^2^	>6	-	-	47.2% (3326)
France	Ducatez et al. [29]	2011–2014	RT-PCR ^3^	ND ^4^	-	-	4.5% (134)
Italy	Rosignoli et al. [3]	2015	IH ^2^	ND ^4^	-	-	92.4% (420)
Italy	Chiapponi et al. [30]	2014–2015	RT-PCR ^3^	ND ^4^	1.3% (150)	-	-
Italy	Rosignoli et al. [3]	2014–2016	RT-PCR ^3^	ND ^4^	8.0% (603)	3.4% (292)	6.5% (895)
Italy	Chiapponi et al. [31]	2018–2019	RT-PCR ^3^	ND ^4^	10.6% (936)	-	-
Luxembourg	Snoeck et al. [32]	2012–2016	IH ^2^	>6	-	80.2% (450)	-
Ireland	O’Donovan et al. [33]	2016–2017	IH ^2^	ND ^4^, >6	64.9% (1183)	94.6% (1219)	-
Ireland	Flynn et al. [19]	2014–2016	RT-PCR ^3^	ND ^4^	5.6% (320)	-	-
U.K.	Dane et al. [34]	2017–2018	RT-PCR ^3^	ND ^4^	8.7% (104)	-	-
China	Jiang et al. [35]	2014	RT-PCR ^3^	ND ^4^	-	0.7% (453)	-
China	Zhai et al. [36]	2016	RT-PCR ^3^	>6	9.7% (404)	1.0% (100)	7.9% (504)
Japan	Horimoto et al. [37]	2010–2016	IH ^2^	>6	-	30.5% (1267)	-
Japan	Murakami et al. [38]	2016	IH ^2^	>6	-	28.6% (28)	-
Japan	Mekata et al. [39]	2016–2017	RT-PCR ^3^	ND ^4^	1.7% (172)	2.4% (205)	2.1% (377)
Japan	Hayakawa et al. [40]	2009–2018	IH ^2^	>6	-	-	57.0% (960)
Turkey	Yilmaz et al. [41]	2018–2019	RT-PCR ^3^	≤6	4.0% (76)	-	-
Togo	Salem et al. [42]	2009, 2015	IH ^2^	ND ^4^	-	-	10.4% (201)
Togo	Fusade-Boyer et al. [43]	2017–2019	IH ^2^	ND ^4^	-	-	4.5% (399)
Marrocco	Salem et al. [42]	2012–2015	IH ^2^	ND ^4^	-	-	35.0% (200)
Benin	Salem et al. [42]	2012, 2014	IH ^2^	ND ^4^	-	-	1.9% (207)

^1^ BRD, bovine respiratory disease. ^2^ IH, hemagglutination inhibition. ^3^ RT-PCR, real time reverse transcription polymerase chain reaction. ^4^ ND, not determined.

## Data Availability

Not applicable.

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
