# Peer review of "Influenza D Virus: A Review and Update of Its Role in Bovine Respiratory Syndrome"

_viruses, 2022, doi:10.3390/v14122717_

Round 1
Reviewer 1 Report
Dear authors,
This is a very comprehensive manuscript that I found very useful for scientific society, putting together all relevant facts on IDV. I have just few minor comments:
Line 184: Originally, reference 9 was about the feral pigs, so I suggest replacing wild boar with feral pigs.
3.6. Diagnosis: I suggest authors to discuss/recommend the possibility of IDV inclusion in multiplex PCR for BRD. Also, the authors should add based on which genome part the genetic lineages are distinctive.
Author Response
Dear Reviewer,
Please, see file attached.
Sincerelly yours,

Reviewer 2 Report
The review on Influenza D virus provides a nice overview of the current knowledge on this virus that is relatively recently introduced in cattle. The review is nicely written, some small typo’s, and I missed a discussion of the virus being able to replicate on human cells in section 3.8. See point by point comments below
Line 63: “the” can be deleted
Line 64: “IDV virus”: the word “virus” can be deleted
Line 145: is there a reference number missing after “…..respiratory disease.”?
Line 204: is there a reference number missing after “….. (HI) assay”?
Line 243: Reference 37 has several (co-)authors, not only one
Line 359: “it” can be deleted
Line 426: “, ,” is there a word missing?
Line 570: “how” can be deleted, and also the rest of the sentence is not very smooth
Section 3.8: IDV can efficiently replicate in human airway epithelia. This finding by Holwerda et al is missing in the review. (see Viruses 2019, 11(4), 377; https://doi.org/10.3390/v11040377)
Author Response
Dear Reviewer,
Please, see atached file.
Sincerelly yours,
